

# Deep learning model for gastrointestinal polyp segmentation

Zitong Wang[1], Zeyi Wang[2] and Pengyu Sun[3]

[1] Imperial College London, London, South Kensington, United Kingdom
[2] Queen Mary University of London, London, Bethnal Green, United Kingdom
[3] Xidian University, Xi'an, Shannxi, China

## ABSTRACT

One of the biggest hazards to cancer-related mortality globally is colorectal cancer, and improved patient outcomes are greatly influenced by early identification. Colonoscopy is a highly effective screening method, yet segmentation and detection remain challenging aspects due to the heterogeneity and variability of readers' interpretations of polyps. In this work, we introduce a novel deep learning architecture for gastrointestinal polyp segmentation in the Kvasir-SEG dataset. Our method employs an encoder-decoder structure with a pre-trained ConvNeXt model as the encoder to learn multi-scale feature representations. The feature maps are passed through a ConvNeXt Block and then through a decoder network consisting of three decoder blocks. Our key contribution is the employment of a cross-attention mechanism that creates shortcut connections between the decoder and encoder to maximize feature retention and reduce information loss. In addition, we introduce a Residual Transformer Block in the decoder that learns long-term dependency by using self-attention mechanisms and enhance feature representations. We evaluate our model on the Kvasir-SEG dataset, achieving a Dice coefficient of 0.8715 and mean intersection over union (mIoU) of 0.8021. Our methodology demonstrates state-of-the-art performance in gastrointestinal polyp segmentation and its feasibility of being used as part of clinical pipelines to assist with automated detection and diagnosis of polyps.

## INTRODUCTION

Colorectal cancer is a significant worldwide health concern with a high mortality rate and a complex course of development which typically starts in benign polyps. Polyps can become precancerous lesions, for which reason there is the utmost need for early discovery and intervention. Effective screening and early polyp removal through colonoscopy are able to mitigate the severity of colorectal cancer, ultimately decreasing its incidence and mortality rates (*Torre et al., 2015*; *Song et al., 2020*). The American Cancer Society emphasizes the fact that normal screening can be able to pick up polyps before they turn into cancer, emphasizing the relevance of preventive intervention in the case of colorectal health management (*Torre et al., 2015*).

Risk factors associated with colorectal cancer be broadly divided into modifiable and non-modifiable factors. Obesity, physical inactivity, poor dietary habits, and excessive

Corresponding author
Zitong Wang, wangzt_uk@163.com

alcohol consumption are modifiable risk factors, each of which is linked to a higher risk of the development of colorectal polyps and cancer (*Um et al., 2020*). Age, cancer or polyp family or personal history, and genetic syndromes are non-modifiable risk factors, all of which put an individual at risk of colorectal neoplasms. Identifying these risk factors is central to the design of targeted screening methods and public health interventions towards alleviating the burden of colorectal cancer.

Despite improvements in screening technology, detection of polyps remains difficult during colonoscopy examinations. Existence of blind spots and human mistake can significantly impact the adenoma detection rate, which is an important measure of colonoscopy efficacy (*Li et al., 2023*). According to reports in this study, the rate of missed polyps ranges from 22% to 28%, which makes it necessary to improve the detection processes. Polyp segmentation is very important. It has a direct influence on the clinical management of colorectal cancer. Precise segmentation allows for better polyp parameter and morphology evaluation, which are very significant during the planning of resection methods and follow-up treatment (*Su et al., 2021*).

In this regard, artificial intelligence (AI) is appearing as a feasible solution to enhance polyp detection and segmentation. AI-driven systems are capable of analyzing colonoscopic images with higher accuracy and speed and therefore minimize the number of false negatives along with enhanced overall diagnostic outcomes (*Kang & Gwak, 2019*; *Singstad & Tzavara, 2021*). In addition, advances in segmentation techniques, such as the utilization of modified U-Net architectures and the use of attention mechanisms, have achieved greater performance in identifying polyp edges and thereby improving diagnostic accuracy (*Yang & Cui, 2024*; *Fu et al., 2022*). With the ever-growing sophistication of medical imaging, utilizing AI and machine learning for polyp segmentation is a basic step toward more effective approaches to the prevention of colorectal cancer.

Endoscopy AI-based decision support systems have played very important roles in augmenting the ability of doctors to identify and segment polyps. The systems leverage sophisticated image processing and segmentation techniques, which are important to ensure that colonoscopy-based polyp detection is effective and efficient. Their central aim is to help clinicians decide evidence-based for the segmentation of polyps, thereby addressing the problem of the polyp miss rate at the highest of 26% for small adenomas (*Yeung et al., 2021*). With assisted deep learning architectures like U-Net and its variants, these systems are able to carry out real-time polyp detection and segmentation, offering a solid tool to gastroenterologists (*Ahmad et al., 2019*; *Chen, Urban & Baldi, 2022*). For example, a study by *Kang & Gwak (2019)* has proven the application of ensemble models to polyp segmentation using methods such as fuzzy clustering and machine learning classifiers to optimize detection capability. Moreover, advances in convolutional neural networks (CNNs) have shown promise for automating segmentation to facilitate faster and more accurate detection of polyps throughout a colonoscopy (*Hossain et al., 2023*; *Guo et al., 2019*). These innovations aim not only to raise detection rates but also reduce the time spent on endoscopic procedures, eventually contributing to improved patient care (*Kang & Gwak, 2019*; *Li et al., 2023*).

Recent studies have established that AI-assisted colonoscopy not only enhances the rate of polyp detection of colorectal polyps but also enhances endoscopic procedure overall diagnostic accuracy. For example, applying deep learning algorithms has demonstrated spectacular success in segmenting and detecting polyps from static images and video streams recorded during colonoscopy (*Kavitha et al., 2022*; *Lee et al., 2020*). These improvements are particularly beneficial, given the difficult-to-view visual characteristics of polyps in conventional methods are more likely to fail and miss during inspection (*Wang et al., 2024b*; *Ramzan et al., 2022*). AI algorithms are configured to operate independently of the clinician's knowledge, which is a reproducible and standard method of identifying the polyps easily missed (*Yamada et al., 2019*; *Khalaf, Rizkala & Repici, 2024*).

Further, AI-driven decision support systems provide more than just detection to offer classification and characterization of polyps, which is crucial in suggesting the right clinical management plans. For instance, CNN-based systems have accurately classified polyps into various categories to facilitate the decision on resecting or not resecting, or following the lesions (*Ozawa et al., 2020*; *García-Rodríguez et al., 2022*). This capability is key to the application of techniques such as "resect and discard" that depend on successful *in vivo* differentiation of polyps for optimal outcomes in patients (*Rao et al., 2022*; *Ahmad et al., 2019*). Thus, the integration of AI into endoscopic practice is a paradigm shift toward more precise and personalized patient management in the prevention and treatment of colorectal cancer.

**Motivation:** AI-based endoscopy decision support systems utilize sophisticated image processing and segmentation techniques to enhance polyp detection and support clinicians in making important decisions regarding polyp management. The systems are aimed at reducing the rate of missing polyps while collaborating to increase the overall quality of colonoscopy, eventually translating to better patient outcomes in the screening and prevention of colorectal cancer (*Mehta et al., 2023*; *Taghiakbari, Mori & von Renteln, 2021*). With this technology still developing, its potential to transform endoscopic procedures and enhance clinical decision-making remains substantial, calling for continued research and development in this field (*Peng et al., 2024*; *Tham et al., 2023*).

In this study, we present a novel deep learning framework for gastrointestinal polyp segmentation that draws upon recent advances in computer vision and transformer-based neural networks. Our proposed model, based on a ConvNeXt (*Liu et al., 2022*) encoder and residual transformer blocks for the decoder, is designed to overcome the limitations of existing approaches by improving feature extraction, maintaining spatial information, and maximizing computational efficiency. With the addition of cross-attention mechanisms and multi-scale feature fusion, our strategy is designed to yield enhanced segmentation accuracy. The purpose of this study is to bridge the gap between clinical usefulness and research advancements, providing a robust and generalizable solution to automated polyp segmentation in endoscopic imaging.

## RELATED WORK

Segmentation and localisation of polyps in medical images, particularly colonoscopy, is one area that holds extreme significance in research due to its ability for early detection and treatment of colorectal cancer. This field has seen tremendous advancements in recent years with the help of various methodologies ranging from basic image processing methods to sophisticated deep learning approaches.

In 2020, *Mandal & Chaudhuri (2020)* introduced a polyp segmentation method based on fuzzy clustering that achieved a high accuracy of 98.80%. The method demonstrated the effectiveness of fuzzy clustering in distinguishing polyps from other tissues, showing the capability of traditional algorithms to achieve high performance in medical image segmentation. The effectiveness of fuzzy clustering in this case agrees with other studies on the subject that show its accuracy in handling uncertainties that are typical in medical images (*Kembaren, Sitompul & Sawaluddin, 2022*). However, even though classical methods like fuzzy clustering are promising, they have poor performance under the advanced variability in polyp appearances, and hence the need to explore more advanced techniques. The following year, *Jha et al. (2021)* employed deep learning architectures in the form of ResUNet++ to enhance polyp segmentation accuracy using data augmentation and Conditional Random Field techniques. Their study emphasized the importance of integrating machine learning with traditional image processing methods to improve segmentation outcomes. The pairing of CRF with deep learning models has been demonstrated to improve boundary delineation, the ability to accurately separate polyps from the surrounding mucosa is one of the most significant issues in polyp segmentation. This merging of methods is part of a larger trend in the literature, as hybrid models are becoming more popular due to their better performance on difficult segmentation tasks. *Banik et al. (2021)* also advanced the field by introducing Polyp-Net, a fusion-based network for segmentation that improved on previous methods in accuracy. Their research illustrated the effectiveness of combining various feature extraction approaches to optimize segmentation accuracy. This was with the work of *Zhou & Li (2023)*, who introduced a hybrid spatial-channel attention and global-regional context aggregation feature to improve segmentation, whose mean Dice score was 0.915. The emphasis on multi-scale integration of features is particularly relevant because polyps tend to be quite variable in shape and size and thus need models that can process these variations with ease. In 2022, *Tran et al. (2022)* introduced the MRR-UNet model, which achieved minimal model size while obtaining an average Dice score of 93.54%. This model is a case of the ongoing effort to balance performance with computational efficiency, a crucial factor in healthcare contexts where real-time processing usually necessary. The trend of optimizing model structures for accuracy and efficiency is seen in the literature, where scientists are increasingly interested in developing lightweight models with high performance levels (*Jha et al., 2021*). Besides, *Mohapatra et al. (2022)* proposed U-PolySeg, which effectively combined features to achieve high accuracy in polyp segmentation.

Despite these advancements, their reliable and precise segmentation is difficult to provide on a consistent basis across large data sets and disease states. Heterogeneity of

polyp texture, variability of size and shape, contribute to this process (*Zhou & Li, 2023*). Moreover, in colonoscopy images, presence of artifact as well as noise makes precise segmentation even more challenging to attain (*Zhang et al., 2024*). As pointed out by *Ji et al. (2024)*, nearly a quarter of polyps in the tumor go unseen during colonoscopy examinations, which underlines the need for robust segmentation techniques that can assist clinicians in real-time. In order to address these challenges, integrating multi-task learning paradigms, as suggested by *Xu et al. (2024)*, might enable models to learn from similar tasks simultaneously, thereby becoming more capable of generalizing across a wide range of polyp appearances and imaging conditions. The study of transfer learning techniques, as observed by *Singstad & Tzavara (2021)*, can provide a way of improving segmentation performance, particularly where there is not much labeled data. By employing pre-trained models on large data, researchers are able to fine-tune the models for specific polyp segmentation tasks, which could lead to improved accuracy and efficiency.

# MEDICAL IMAGE SEGMENTATION

In computer-aided diagnosis, medical image segmentation is a crucial technique that allows for accurate anatomical structure boundary delineation. The purpose of segmentation is to separate a medical image into various regions corresponding to different tissue types, organs, or pathologic regions, upon which one can perform more accurate medical imaging analysis and decision-making. An example is shown in Fig. 1.

Mathematically, a common method to formulate medical image segmentation is as a pixel-wise classification task. Each pixel $(x, y)$ in the image domain $\Omega$ is given a label $L(x, y)$ *via* segmentation given an input medical image $I$, such that:

$$L : \Omega \rightarrow \{0, 1, \ldots, K - 1\}, \tag{1}$$

where $K$ is the number of classes, and $L(x, y)$ denotes the assigned class label.

Segmentation methods can be broadly categorized into two main approaches.

## Image processing-based segmentation

Traditional methods rely on thresholding, edge detection, and region growing techniques to partition the image. For example, Otsu's thresholding method (*Otsu, 1979*) selects an optimal threshold $T$ by minimizing intra-class variance:

$$T = \arg \min_{T} \left[ \sigma_1^2(T) + \sigma_2^2(T) \right], \tag{2}$$

where $\sigma_1^2$ and $\sigma_2^2$ are the variances of pixel intensities for background and foreground regions, respectively.

## Machine learning-based segmentation

Machine learning-based medical image segmentation has long relied on classical approaches that leverage handcrafted features and statistical learning techniques to delineate anatomical structures or pathological regions. Traditional methods such as

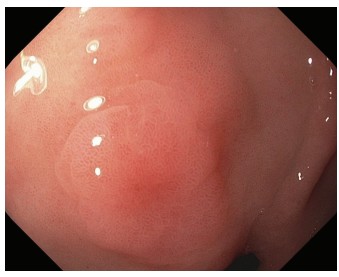 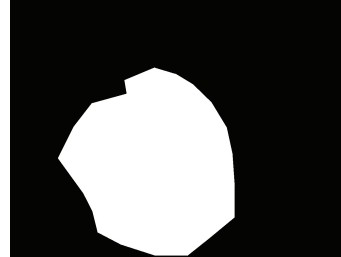

| Original image | Ground Truth |

**Figure 1  An example of segmentation task.**

support vector machines, k-nearest neighbours, random forests, and clustering algorithms (*Cai et al., 2015*) have been widely used due to their interpretability and relatively low computational requirements. These methods typically operate on extracted features–such as texture, intensity, or shape descriptors–from pixel or superpixel regions to classify each region into predefined categories. More advanced approaches incorporate kernel methods to capture non-linear patterns in the feature space, improving segmentation performance in complex imaging scenarios. In particular, kernel sparse representation (KSR) has shown promise by mapping image data into high-dimensional feature spaces, where sparse coding techniques can better capture the underlying structure and discriminative information. KSR-based segmentation methods provide a flexible framework that balances sparsity and kernel-induced nonlinearity (*Chen et al., 2017a*), making them suitable for challenging tasks such as tumor boundary delineation or tissue classification in modalities like MRI and CT (*Chen et al., 2017b*).

While classical machine learning methods have provided valuable insights and reasonable performance in medical image segmentation, they often depend heavily on handcrafted features and may struggle with complex or high-dimensional data. To overcome these limitations, deep learning-based approaches, particularly CNNs, have emerged as a powerful alternative. CNNs are capable of automatically learning hierarchical and spatially invariant features directly from raw image data, enabling more accurate and robust segmentation across a wide range of medical imaging modalities. CNN-based image segmentation is essential for many medical applications (*Nguyen et al., 2022*), such as disease classification, lesion detection, and prognosis prediction, as it enables precise localisation and delineation of relevant anatomical and pathological structures. One of the notable CNN-based image segmentation approaches is the U-Net (*Ronneberger, Fischer & Brox, 2015*) architecture, which uses employs a skip-connection encoder-decoder scheme. Given an input image $I$, the CNN learns a mapping function $F$ from the image to a segmented output $\hat{L}$:

$$\hat{L} = F(I; \theta), \tag{3}$$

where $\theta$ represents the network parameters learned during training.

Loss functions in deep learning segmentation models often combine Dice loss and cross-entropy loss, defined as:

$$\mathscr{L}_{Dice} = 1 - \frac{2\sum_i p_i g_i}{\sum_i p_i^2 + \sum_i g_i^2},$$

(4)

where $p_i$ and $g_i$ represent predicted and ground truth segmentation masks, respectively.

# MATERIALS AND METHOD

## The Kvasir-SEG dataset

In this article, we utilize the Kvasir-SEG dataset (*Jha et al., 2019a*), a publicly available gastrointestinal polyp segmentation dataset. The dataset consists of 1,000 high-quality polyp images from real colonoscopy procedures and their corresponding pixel-wise ground truth masks. The images are all annotated by experienced medical physicians to ensure accuracy in segmentation tasks. The dataset consists of a diverse range of polyp shapes, sizes, and textures, making it extremely well-suited as a training and test set for deep learning networks in analysis of medical images. Kvasir-SEG is a valuable benchmark for building robust and generalizable polyp segmentation techniques and opens the door for computer-aided diagnosis of gastrointestinal disorders.

### Data processing

We used image augmentation techniques such as random cropping, flipping, scaling, rotation, cutoff, brightness, and random erasing to increase our training dataset in order to improve the durability and accuracy of deep learning models for polyp segmentation. These augmentation techniques facilitate the enhancement of the diversity of the polyp samples in the Kvasir-SEG dataset, ultimately leading to improved segmentation accuracy and generalization in real-world settings. Additionally, intensity transformations like contrast adjustments and Gaussian noise addition were employed to improve image diversity. Finally, scale augmentation techniques, such as rescaling images to different resolutions, further increased the robustness of the trained models. After all preprocessing was completed, images were resized to $320 \times 320$ pixels. Preprocessing augmentation (applied before training) has been leveraged to enhance model robustness and reduce overfitting, ultimately leading to improved segmentation accuracy and generalization in real-world scenarios. In terms of model training and testing, 80% of the dataset was used for training, 10% for validation and 10% for testing.

## Proposed method

In this study, we present our architecture in Fig. 2. This model operates as an encoder-decoder network, commencing with a pre-trained ConvNeXt model serving as the encoder. This pre-trained encoder receives the input image to extract three distinct intermediate feature maps. These feature maps undergo processing through a ConvNeXt Block, which is shown in the next sections.

The subsequent component is the decoder network, consisting of three decoder blocks. The first decoder block receives the reduced feature map, which first passes through an upsampling layer that increases the spatial dimensions. The upsampled feature map is then

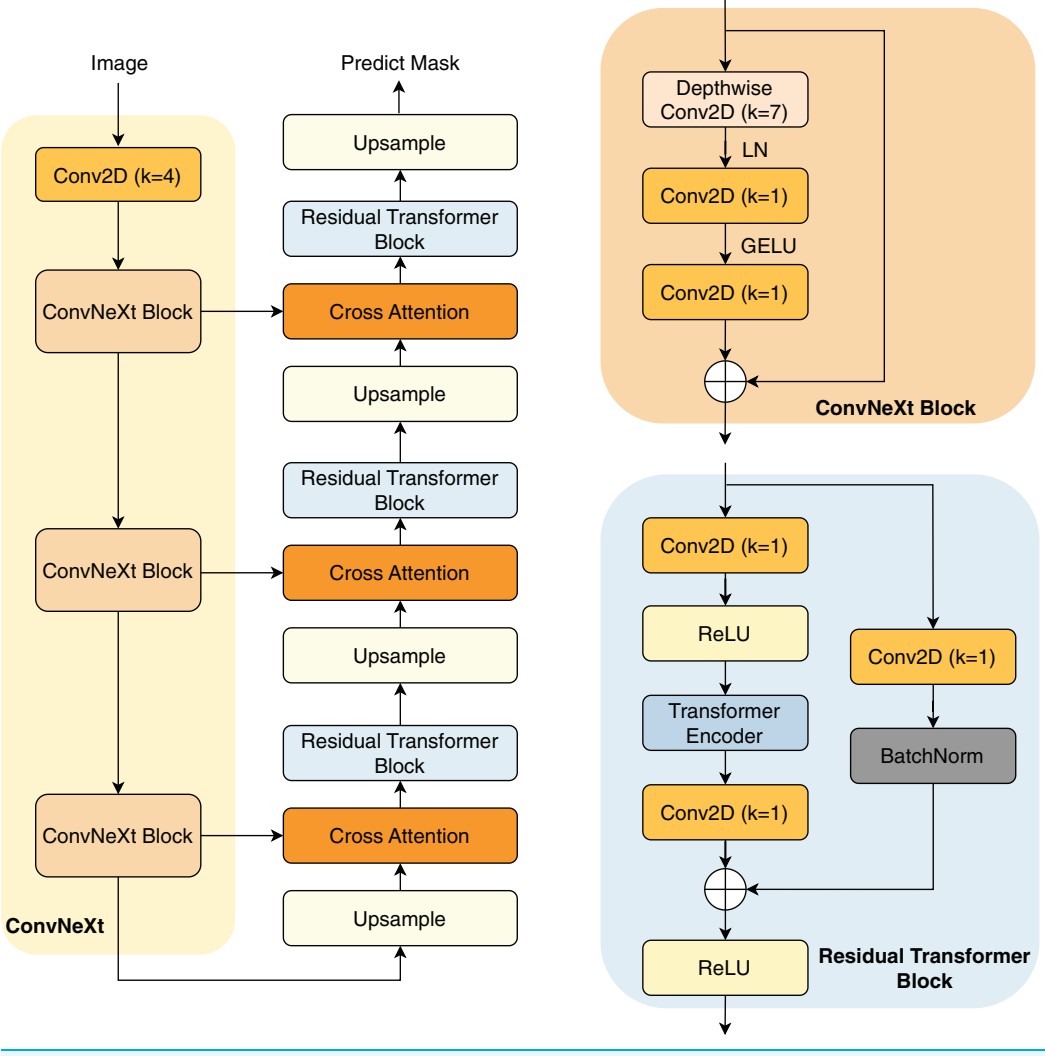

**Figure 2 Overview of our architecture.**

combined with another reduced feature map of matching spatial dimensions, establishing a cross attention block to create a shortcut connection between the encoder and the decoder. This connection facilitates improved information flow, mitigating potential feature loss due to the network's depth. Furthermore, the combination of feature maps through an attention layer helps the model learn important characteristics between the features, thereby improving their performance.

The combined feature maps are processed through our proposed residual transformer block. In this block, the feature maps are reshaped into patches before being input into the transformer layers (*Vaswani et al., 2017*), which include multi-head self-attention mechanisms for enhanced feature representation learning. Notably, in the final decoder block, the residual transformer block is substituted with a simpler residual block to minimize trainable parameters; the architecture of the residual transformer block is described in the next section. The output from the final decoder goes through an

upsampling layer, before being processed by a convolution layer with kernel $1 \times 1$ (Conv2D with $k = 1$) and a sigmoid activation function to predict the mask.

### ConvNeXt block

The ConvNeXt Block starts with a depthwise $7 \times 7$ convolution, expanding the receptive field while preserving spatial structures. This is followed by layer normalization (LN) instead of batch normalization, improving stability and efficiency. The normalized features are then processed through a pointwise Conv2D ($k = 1$), which expands the channel dimension similarly to the inverted bottleneck structure in MobileNetV2 (*Sandler et al., 2018*). A GELU activation function is applied before another Conv2D ($k = 1$), which restores the original channel dimension. To enhance expressiveness, a stochastic depth mechanism may be employed. Finally, the processed feature map is added back to the input *via* a residual connection, maintaining gradient flow and improving convergence. This streamlined design, inspired by modern transformer architectures, improves performance and efficiency while retaining the hierarchical nature of convolutional networks.

### Residual transformer block

A convolution layer with kernel $1 \times 1$ (Conv2D with $k = 1$) is used to start the residual transformer block. Batch normalization and a ReLU activation function come next. The feature maps are then flattened, maintaining a constant patch size of four. These flattened maps are directed into the transformer block, which comprises four heads and two layers. The transformer block performs self-attention on the feature maps, enhancing the network's robustness. The output of the transformer block is reshaped back to the original input dimensions. Following this, the feature map undergoes another Conv2D ($k = 1$), and is then added to the input feature maps before being processed through the ReLU activation function.

### Cross attention

Cross-attention (*Chen, Fan & Panda, 2021*) is a mechanism where there are two input sequences: a query sequence $Q$ and a key-value sequence $(K, V)$, where $K$ and $V$ typically come from a different source than $Q$. As shown in Fig. 3, input 1 corresponds to the features obtained from the residual transformer block, and input 2 corresponds to the features of the ConvNeXt Block. Dot-product attention is computed between the query $Q$ and the key $K$:

$$\text{Attention scores} = \frac{QK^T}{\sqrt{d_k}}, \tag{5}$$

where:

- $Q$ is the query matrix of shape $(n_q, d_k)$, where $n_q$ is the number of queries and $d_k$ is the dimensionality of the key.
- $K$ is the key matrix of shape $(n_k, d_k)$, where $n_k$ is the number of keys.
- The dot product is scaled by $\sqrt{d_k}$ to prevent large values when $d_k$ is large.

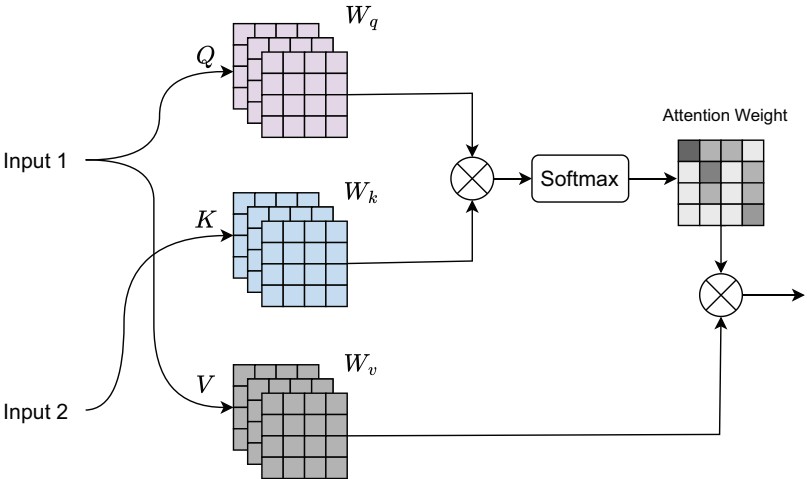

**Figure 3 Cross attention visualization.**

Softmax normalization is applied to the attention scores to obtain the attention weights:

$$\text{Attention weights} = \text{Softmax}\left(\frac{QK^T}{\sqrt{d_k}}\right). \tag{6}$$

The softmax ensures that the attention weights are between 0 and 1, and sum to 1. Weighted sum of the values $V$ is computed using the attention weights:

$$\text{Output} = \text{Attention Weights} \times V, \tag{7}$$

where $V$ is the value matrix of shape $(n_k, d_v)$, with $d_v$ being the dimensionality of the value.

## Evaluation metrics

The performance of various architectures for medical image segmentation tasks can be evaluated and compared using multiple metrics. In this study, to assess segmentation performance, we calculate the Dice coefficient, mean IoU, recall, and precision based on this classification.

**Dice coefficient** is a widely used metric for measuring the similarity between ground truth and anticipated segmentation at the pixel level. It is defined as:

$$Dice(A, B) = \frac{2 \times |A \cap B|}{|A| + |B|} = \frac{2 \times TP}{2 \times TP + FP + FN}, \tag{8}$$

where:

- $A$ represents the set of predicted pixels,
- $B$ represents the ground truth pixels,
- $TP$ (true positive) denotes correctly predicted polyp pixels,
- $FP$ (false positive) refers to incorrectly predicted polyp pixels,
- $FN$ (false negative) represents polyp pixels that were not correctly detected.

**Intersection over union (IoU)** is another common measure for assessing segmentation accuracy is the Jaccard index. The overlap between the ground truth ($B$) and the anticipated segmentation ($A$) is measured as follows:

$$IoU(A, B) = \frac{|A \cap B|}{|A \cup B|} = \frac{TP}{TP + FP + FN} \tag{9}$$

**Recall and precision** are defined as follows:

$$Recall = \frac{TP}{TP + FN} \tag{10}$$

$$Precision = \frac{TP}{TP + FP} \tag{11}$$

# EXPERIMENTAL RESULTS AND DISCUSSION

## Setup

In our experiments, we implemented methods using the PyTorch and conducted training on a single NVIDIA RTX 3060 GPU with 12 GB RAM. The AdamW optimizer (*Loshchilov & Hutter, 2017*) was used for 50 epochs of training with a batch size of 16, a learning rate of 0.0001 and a weight decay of 1e−4. CUDA 14.2 and Windows 11 were installed in the software environment.

## Comparing with benchmarking models

To conduct a comprehensive performance analysis, we compared our model with several advanced deep learning models for gastrointestinal polyp segmentation. These models include:

- **U-Net** (*Ronneberger, Fischer & Brox, 2015*): An encoder-decoder network with skip connections is utilized by U-Net in biomedical image segmentation to enhance feature extraction and reconstruct polyp shapes more effectively.
- **U-Net++** (*Zhou et al., 2018*): An extension of U-Net, the model incorporates a nested and dense skip connection design to yield a more refined feature representation and improved segmentation accuracy.
- **ResU-Net++** (*Jha et al., 2019b*): The network consists of residual links and squeeze-and-excitation units to facilitate feature propagation and obtain valuable contextual information but is poorer on lower segmentation precision.
- **UACANet** (*Kim, Lee & Kim, 2021*): A state-of-the-art attention-based model with uncertainty-aware learning to enhance segmentation results and yield higher Dice similarity coefficients (DSC).
- **UNeXt** (*Valanarasu & Patel, 2022*): This architecture employs a transformer-based approach towards medical image segmentation, employing axial attention mechanisms in order to better represent features.
- **HarDNet-MSEG** (*Huang, Wu & Lin, 2021*): Based on the HarDNet backbone, this model optimizes computational efficiency while maintaining competitive performance in polyp segmentation tasks.

- **ColonSegNet** (*Jha et al., 2021*): A domain-specific model specially developed for the task of colon polyp segmentation, employing the use of multi-scale attention so as to perform better detection.
- **SegFormer** (*Xie et al., 2021*): SegFormer is a lightweight yet powerful semantic segmentation framework that combines a hierarchical Transformer encoder with a simple, MLP-based decoder, eliminating the need for complex post-processing. Its strong generalization ability and efficient architecture make it well-suited for medical image segmentation, where both accuracy and computational efficiency are critical.
- **Swin Transformer** (*Liu et al., 2021*): introduces a hierarchical vision architecture with shifted windows, enabling efficient modeling of both local and global context, an essential capability for medical image segmentation where subtle anatomical structures must be captured with high precision.

The results in Table 1 demonstrate the superior performance of our proposed model on the Kvasir-SEG test dataset across all key evaluation metrics. Our method achieves the highest mIoU of 0.8021 and DSC of 0.8659, outperforming recent state-of-the-art models such as SegFormer (mIoU = 0.7872, DSC = 0.8567) and Swin Transformer (mIoU = 0.7754, DSC = 0.8502).

Notably, our model also achieves the highest precision (0.9023) and recall (0.8796), indicating a strong balance between detecting true polyp regions and minimizing false positives. This is particularly significant in the medical domain, where both missed detections and over-segmentation can have critical clinical implications.

Compared to earlier CNN-based methods such as U-Net and U-Net++, our approach shows a marked improvement, reflecting the benefit of incorporating advanced components like the ConvNeXt encoder, cross-attention mechanisms, and Residual Transformer Blocks. Even against transformer-based architectures like SegFormer and Swin Transformer, our model achieves a better trade-off between segmentation accuracy and boundary refinement, making it a compelling choice for automated polyp detection systems in clinical endoscopy workflows.

## Ablation study

To assess the contributions of key architectural components in our proposed polyp segmentation model, we conducted a comprehensive ablation study. This study evaluates the effects of different backbone networks and the effectiveness of the cross attention mechanism in enhancing segmentation performance. By systematically varying these components, we aim to quantify their influence on key metrics such as mIoU, DSC, recall, and precision. The following sections present our findings, highlighting the advantages of ConvNeXt as the feature extractor and the role of cross attention in refining segmentation accuracy.

### *Impact of the ConvNeXt backbone on segmentation performance*

To assess the efficacy of the ConvNeXt backbone in our proposed polyp segmentation model, we conducted an ablation study comparing it against various pretrained feature

**Table 1 Performance results on the Kvasir-SEG test dataset.**

| Method | mIoU | DSC | Recall | Precision |
|---|---|---|---|---|
| U-Net (*Ronneberger, Fischer & Brox, 2015*) | 0.7423 | 0.8215 | 0.8467 | 0.8654 |
| U-Net++ (*Zhou et al., 2018*) | 0.7352 | 0.8160 | 0.8374 | 0.8558 |
| ResU-Net++ (*Jha et al., 2019a*) | 0.5258 | 0.6354 | 0.6879 | 0.7041 |
| UACANet (*Kim, Lee & Kim, 2021*) | 0.7648 | 0.8462 | 0.8743 | 0.8657 |
| HarDNet-MSEG (*Huang, Wu & Lin, 2021*) | 0.7410 | 0.8206 | 0.8438 | 0.8579 |
| ColonSegNet (*Jha et al., 2021*) | 0.6897 | 0.7852 | 0.8123 | 0.8370 |
| UNeXt (*Valanarasu & Patel, 2022*) | 0.6238 | 0.7281 | 0.7792 | 0.7583 |
| SegFormer (*Xie et al., 2021*) | 0.7872 | 0.8567 | 0.8728 | 0.8910 |
| Swin Transformer (*Liu et al., 2021*) | 0.7754 | 0.8502 | 0.8701 | 0.8845 |
| Ours | **0.8021** | **0.8659** | **0.8796** | **0.9023** |

extractors, including VGG16 (*Simonyan & Zisserman, 2014*), ResNet50 (*He et al., 2015*), ViT (*Dosovitskiy et al., 2020*), InceptionV3 (*Szegedy et al., 2015*), and EfficientNet (*Tan & Le, 2019*). We integrated each of these models into our segmentation framework and assessed their performance on the Kvasir-SEG validation dataset.

The results in Table 2 provide compelling evidence for the superior performance of the ConvNeXt backbone in the context of gastrointestinal polyp segmentation. ConvNeXt consistently outperformed all other evaluated backbones—including well-established CNN architectures such as VGG16 and ResNet50, as well as more recent models like ViT and EfficientNet, across all key metrics: mIoU, DSC, recall, and precision.

Most notably, ConvNeXt achieved a mIoU of 0.8120 and a DSC of 0.8802, significantly higher than the best-performing alternative (EfficientNet: mIoU = 0.7635, DSC = 0.8385). This substantial improvement highlights ConvNeXt's ability to retain spatial details while capturing deep semantic features, which is crucial for accurate boundary delineation in segmentation tasks.

Additionally, ConvNeXt achieved the highest recall (0.8950) and precision (0.9100), indicating its robustness in correctly identifying polyps without over-segmenting non-polyp areas. This balance is particularly important in clinical settings, where both false negatives (missed polyps) and false positives (unnecessary interventions) carry significant risks.

The strong performance of ConvNeXt can be attributed to its hybrid design, which modernizes convolutional structures with transformer-inspired training strategies. Unlike VGG16 and ResNet50, which rely on deeper but relatively rigid convolutional layers, ConvNeXt incorporates design principles such as depthwise convolutions, layer scaling, and expanded kernel sizes that enhance its representational power while maintaining computational efficiency.

In contrast, while ViT brings strong global attention capabilities, its reliance on large-scale pretraining and less inductive bias may hinder performance on limited-medical-data domains like Kvasir-SEG. Similarly, InceptionV3 and EfficientNet, though strong contenders, fall short in balancing fine-grained localization with semantic richness.

**Table 2 Performance comparison of different backbone networks on the Kvasir-SEG validation dataset.**

| Backbone | mIoU | DSC | Recall | Precision |
|---|---|---|---|---|
| VGG16 | 0.7298 | 0.8124 | 0.8340 | 0.8475 |
| ResNet50 | 0.7567 | 0.8332 | 0.8578 | 0.8689 |
| ViT | 0.7463 | 0.8248 | 0.8441 | 0.8612 |
| InceptionV3 | 0.7420 | 0.8215 | 0.8406 | 0.8570 |
| EfficientNet | 0.7635 | 0.8385 | 0.8615 | 0.8710 |
| **Ours** | **0.8120** | **0.8802** | **0.8950** | **0.9100** |

Overall, these findings strongly justify our choice of ConvNeXt as the feature extractor in our model. Its ability to combine precision, recall, and contextual understanding makes it particularly well-suited for the complex visual patterns present in endoscopic polyp images.

### Impact of attention mechanisms on segmentation performance

To evaluate the effectiveness of the cross attention block in our proposed architecture, we carried out an ablation study by comparing different model variants with and without the cross attention mechanism.

The results presented in Table 3 clearly demonstrate the progressive improvements brought by each architectural enhancement, culminating in the superior performance of the full model incorporating the cross-attention mechanism.

Starting with the baseline model that uses a ConvNeXt encoder with standard skip connections, we observe a respectable performance (mIoU = 0.7708, DSC = 0.8478), highlighting the strength of ConvNeXt as a feature extractor. However, replacing the standard decoder with a Transformer-based decoder improves both mIoU and DSC (to 0.7829 and 0.8563, respectively), suggesting that self-attention layers better capture long-range dependencies and contextual information.

The inclusion of a residual transformer block without cross-attention yields further gains (mIoU = 0.7879, DSC = 0.8590), indicating the benefit of deeper and more expressive decoder structures. Introducing dot-product attention (*Luong, Pham & Manning, 2015*) into this architecture enhances the model's ability to fuse encoder and decoder features more effectively, resulting in slightly improved metrics (mIoU = 0.7895, DSC = 0.8611). A more refined variant using scaled dot-product (*Vaswani et al., 2017*) attention shows continued progress (mIoU = 0.7987, DSC = 0.8689), likely due to its improved gradient stability and better attention scaling.

Finally, the full model, which integrates the ConvNeXt encoder, residual transformer blocks, and a tailored cross-attention mechanism, achieves the best performance across all metrics (mIoU = 0.8120, DSC = 0.8802, recall = 0.8950, precision = 0.9100). These results confirm that cross-attention significantly improves the fusion of encoder-decoder features, enabling the model to more accurately delineate polyp boundaries and reduce both false positives and false negatives.

**Table 3 Impact of the cross-attention mechanism on segmentation performance evaluated on the Kvasir-SEG validation dataset.**

| Method | mIoU | DSC | Recall | Precision |
|---|---|---|---|---|
| Baseline (ConvNeXt encoder + Standard skip connections) | 0.7708 | 0.8478 | 0.8640 | 0.8869 |
| ConvNeXt encoder + Transformer decoder (No cross attention) | 0.7829 | 0.8563 | 0.8700 | 0.8982 |
| ConvNeXt encoder + Residual transformer (No cross attention) | 0.7879 | 0.8590 | 0.8732 | 0.9008 |
| ConvNeXt encoder + Residual transformer (Dot-product attention) | 0.7895 | 0.8611 | 0.8748 | 0.9025 |
| ConvNeXt encoder + Residual transformer (Scaled dot-product attention) | 0.7987 | 0.8689 | 0.8812 | 0.9051 |
| Full model (Ours) | **0.8120** | **0.8802** | **0.8950** | **0.9100** |

## Visualization

To better understand the model's segmentation performance, we visualize both the predicted segmentation masks and the corresponding heatmaps of feature activations. Figure 4 presents sample predictions alongside ground truth masks, highlighting the model's precision in defining polyp boundaries. The visualization results indicate that the model performs well in segmenting polyps with clear boundaries and sufficient contrast against the background. However, we observe occasional misclassifications in cases where polyps exhibit low contrast or are partially occluded. Moreover, flat and smaller polyps possess less clear segmentation masks, meaning that they require more refinement for being more attentive to these challenging cases. Based on the inspection of these predictions, we can identify where improvement can be made, like incorporating segmentation attention mechanisms that are more attentive to finer details or merging more context information from surrounding tissues.

## LIMITATIONS AND FUTURE WORK

While our deep learning model demonstrates strong performance in gastrointestinal polyp segmentation, several limitations remain, pointing toward valuable directions for future research and clinical translation.

First, the generalizability of our model is inherently limited by the Kvasir-SEG dataset's size and diversity. Despite being a well-curated resource, it does not encompass the full variability of polyp shapes, textures, lighting conditions, and endoscopic device outputs seen in real-world clinical environments. Similar concerns about domain adaptability have been noted in related segmentation challenges, such as dental plaque recognition in unconstrained imaging scenarios (*Song et al., 2024*) and retinal fundus enhancement tasks (*Jia, Chen & Chi, 2024*). Future work should prioritize the inclusion of more diverse and cross-device datasets and explore data harmonization strategies to improve model robustness across clinical settings.

Secondly, our model shows reduced performance for small or flat polyps that exhibit minimal contrast with adjacent mucosal tissues. This limitation is consistent with challenges observed in other low-contrast medical segmentation tasks, such as skin lesion detection (*Wang et al., 2025*) and bacterial infection targeting in constrained anatomical regions (*Wang et al., 2024a*). To address this, future research could incorporate specialized

| Image | Ground Truth | Predition | Heatmap |
|---|---|---|---|

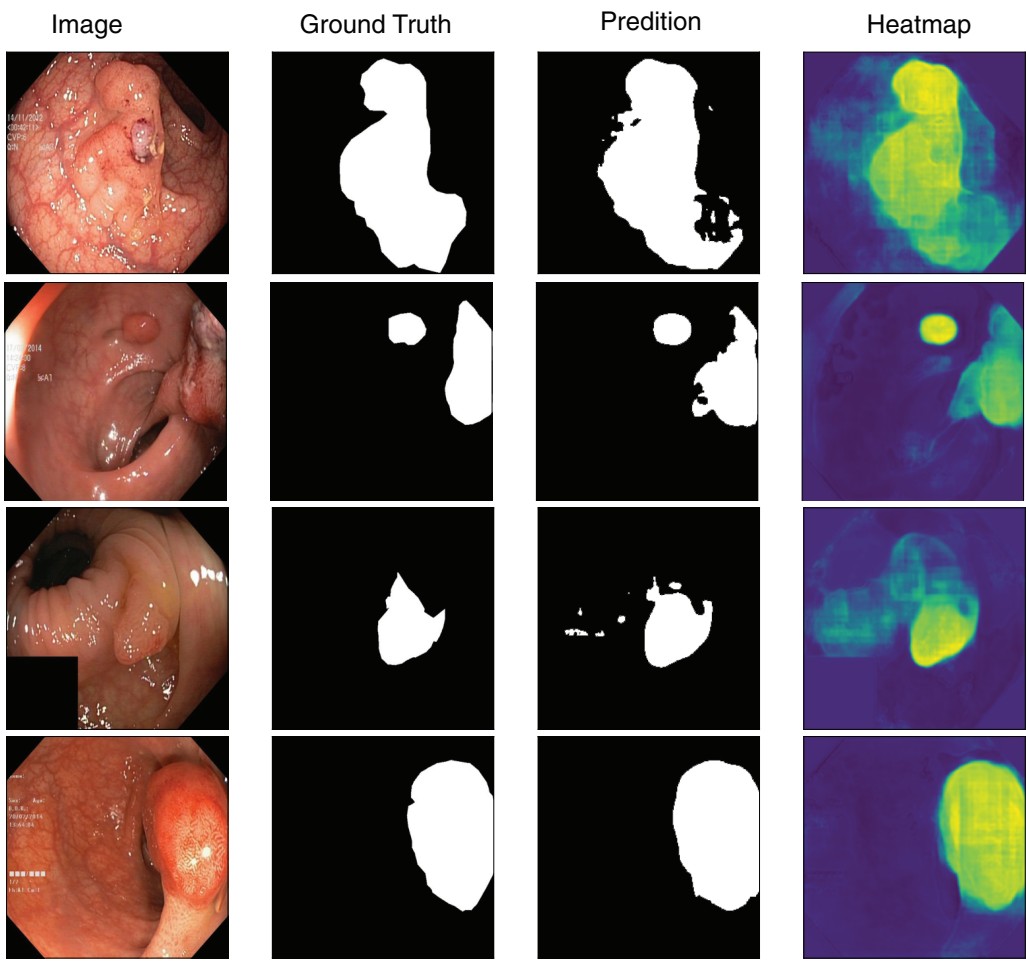

**Figure 4 Visualization of model's prediction for polyp segmentation.**

enhancement techniques—such as contrast-aware modules, domain-specific priors, or generative augmentation—tailored to better distinguish such ambiguous structures.

Additionally, computational complexity remains a barrier to deployment in resource-constrained clinical environments. Although our use of a ConvNeXt encoder and Residual Transformer Blocks significantly improves segmentation accuracy, it also increases inference latency and hardware demands. Insights from fast medical imaging tasks, such as super-resolution ultrasound (*Luan et al., 2023*) and denoising in localization microscopy (*Yu et al., 2023*), highlight the potential of lightweight architectures and real-time optimization. Future work could explore model pruning, quantization, or hybrid encoder-decoder setups that retain accuracy while reducing overhead.

By addressing these limitations through dataset expansion, architectural refinements, and computational efficiency enhancements, future research can bridge the gap between experimental performance and clinical practicality, enabling broader deployment of AI-driven polyp segmentation systems.

## CONCLUSIONS

Utilizing our proposed deep learning model for the segmentation of gastrointestinal polyps, we have established that the use of contemporary techniques such as the ConvNeXt encoder, residual transformer blocks, and cross-attention greatly improves segmentation accuracy. Our method outperforms traditional approaches as evidenced by the achieved Dice score of 0.8715 and mIoU of 0.8021 on the Kvasir-SEG dataset. This success proves the strength of deep learning in overcoming the challenges of polyp detection and segmentation in real-world clinical practice, where efficiency and accuracy are paramount. Moreover, the combining methods of attention with multi-scale feature fusion enables the model to learn more complex patterns in endoscopic images, improving feature extraction and spatial information retention. This work bridges the gap between cutting-edge computer vision research and actual needs of clinical workflows, and it proposes a promising approach to automatic polyp detection and diagnosis in gastrointestinal endoscopy. Our approach provides a strong and efficient tool for assisting medical professionals in the early detection of colorectal cancer, eventually resulting in better patient outcomes and better cancer-related mortality rates worldwide. Future research will concentrate on enhancing the model for application in real-world clinical environments and applying it to other medical imaging tasks.

### Funding

The authors received no funding for this work.

### Competing Interests

The authors declare that they have no competing interests.

### Author Contributions

- Zitong Wang conceived and designed the experiments, performed the experiments, analyzed the data, performed the computation work, prepared figures and/or tables, authored or reviewed drafts of the article, and approved the final draft.
- Zeyi Wang conceived and designed the experiments, performed the experiments, analyzed the data, prepared figures and/or tables, authored or reviewed drafts of the article, and approved the final draft.
- Pengyu Sun conceived and designed the experiments, performed the experiments, analyzed the data, prepared figures and/or tables, authored or reviewed drafts of the article, and approved the final draft.

### Data Availability

The data is sourced from *Jha et al. (2019a)* and is available at Simula: https://datasets.simula.no/kvasir-seg/. The code is available in the Supplemental File.

## Supplemental Information

Supplemental information for this article can be found online at http://dx.doi.org/10.7717/peerj-cs.2924#supplemental-information.

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
