# Peer review of "Deep learning model for gastrointestinal polyp segmentation"

_PeerJ Computer Science, doi:10.7717/peerj-cs.2924_

## Round 0.1 · original submission · Major Revisions

· Academic Editor

Major Revisions

Please revise the manuscript by addressing the comments from both reviewers. Also, consider including more comparisons as suggested by Reviewer 1.

Reviewer 1 ·

Basic reporting

The paper is well written and easy to follow. The methods are clearly explained, and the results are presented in a helpful and organised way. The figures and tables do a good job of showing how the model works and how well it performs. That said, a few parts could use a bit more explanation to give readers the full picture.

Experimental design

The authors mention that they used data augmentation to help train the model, but they did not include any pictures of what these transformations look like. Showing a few examples would make it easier to understand what was done during this step.

ConvNeXt is used as the model’s backbone, which is a strong choice. Still, another well-known model in medical imaging is EfficientNet. It would be helpful to briefly explain why ConvNeXt was preferred in this case.

The ablation study results in Table 3 are interesting, but it is not easy to see the differences just by looking at numbers. A visual version, like a chart or graph, would make the comparisons much clearer.

Validity of the findings

To better support the findings, the authors could try running the model using another architecture, such as the Swin Transformer. Comparing the performance of different backbones would give a fuller picture of how the method stacks up against other options.

Additional comments

It is important to keep the wording consistent throughout the paper. For example, once you define the Dice Similarity Coefficient (DSC), just stick to using “DSC” from then on, rather than repeating the full name. This keeps the writing smooth and avoids repetition.

Cite this review as

Reviewer 2 ·

Basic reporting

- The paper is well written and clearly structured. The authors do a good job explaining the technical details, and the section about related research gives a solid overview of the current work in polyp segmentation. The method is clearly explained, and the results are helpful.
- However, there are a few parts that could be improved to make things even clearer and strengthen the overall study.

Experimental design

- The authors use a cross-attention mechanism in their method. It would be useful to explain why they chose this over other common types of attention mechanisms like dot-product or scaled dot-product attention. Including a short comparison would make the reasoning clearer.
- To better test how strong the method really is, it would also help to compare its performance with newer architectures like SegFormer.

Validity of the findings

- The paper talks about using data augmentation, but it does not show any images to demonstrate what this looks like. Adding a few examples would help readers understand the kinds of changes made to the dataset.
- It would also be good to provide details about each backbone network used, such as how many parameters it has and what input size it expects. This helps readers judge the balance between how complex the models are and how well they perform.
- Finally, Table 3 discusses the effect of the attention mechanism. To make it easier to understand, it would help to include some visuals that show how the different versions of the model behave in the ablation study.

Cite this review as

---

## Round 0.2 · accepted · Accept

· Academic Editor

Accept

The authors have addressed all of the reviewers' comments. Based on reviewers' recommendations and my own assessment, I recommend accepting this manuscript for publication.

Reviewer 1 ·

Basic reporting

All looks good - nothing else to add.

Experimental design

The authors have satisfactorily addressed all revision points, having expanded their experimental validation to include comparisons with contemporary models such as EfficientNet and Swin Transformer. The newly presented results provide convincing support for their proposed method.

Validity of the findings

All of my concerns have been satisfactorily addressed, and I have no further comments.

Additional comments

No further comments.

Cite this review as

Reviewer 2 ·

Basic reporting

The revised version is well-organized and clearly written. I don't have any other comments.

Experimental design

I have no further comments.

Validity of the findings

I have no further comments.

Additional comments

I have no further comments.

Cite this review as